# Extracorporeal Membrane Oxygenation in Congenital Heart Disease

**DOI:** 10.3390/children9030380

**Published:** 2022-03-09

**Authors:** Tanya Perry, Tyler Brown, Andrew Misfeldt, David Lehenbauer, David S. Cooper

**Affiliations:** Heart Institute, Cincinnati Children’s Hospital Medical Center, Cincinnati, OH 45229, USA; tyler.brown@cchmc.org (T.B.); andrew.misfeldt@cchmc.org (A.M.); david.lehenbauer@cchmc.org (D.L.); david.cooper@cchmc.org (D.S.C.)

**Keywords:** ECMO, congenital heart disease, cardiac critical care

## Abstract

Mechanical circulatory support (MCS) is a key therapy in the management of patients with severe cardiac disease or respiratory failure. There are two major forms of MCS commonly employed in the pediatric population—extracorporeal membrane oxygenation (ECMO) and ventricular assist device (VAD). These modalities have overlapping but distinct roles in the management of pediatric patients with severe cardiopulmonary compromise. The use of ECMO to provide circulatory support arose from the development of the first membrane oxygenator by George Clowes in 1957, and subsequent incorporation into pediatric cardiopulmonary bypass (CPB) by Dorson and colleagues. The first successful application of ECMO in children with congenital heart disease undergoing cardiac surgery was reported by Baffes et al. in 1970. For the ensuing nearly two decades, ECMO was performed sparingly and only in specialized centers with varying degrees of success. The formation of the Extracorporeal Life Support Organization (ELSO) in 1989 allowed for the collation of ECMO-related data across multiple centers for the first time. This facilitated development of consensus guidelines for the use of ECMO in various populations. Coupled with improving ECMO technology, these advances resulted in significant improvements in ECMO utilization, morbidity, and mortality. This article will review the use of ECMO in children with congenital heart disease.

## 1. Introduction

### 1.1. History

Mechanical circulatory support (MCS) is a key therapy in the management of patients with severe cardiac disease or respiratory failure [1]. There are two major forms of MCS commonly employed in the pediatric population—extracorporeal membrane oxygenation (ECMO) and ventricular assist device (VAD). These modalities have overlapping but distinct roles in the management of pediatric patients with severe cardiopulmonary compromise. The use of ECMO to provide circulatory support arose from the development of the first membrane oxygenator by George Clowes in 1957, and subsequent incorporation into pediatric cardiopulmonary bypass (CPB) by Dorson and colleagues [2,3]. The first successful application of ECMO in children with congenital heart disease undergoing cardiac surgery was reported by Baffes et al. in 1970 [4]. For the ensuing nearly two decades, ECMO was performed sparingly and only in specialized centers with varying degrees of success. The formation of the Extracorporeal Life Support Organization (ELSO) in 1989 allowed for the collation of ECMO-related data across multiple centers for the first time [5]. This facilitated development of consensus guidelines for the use of ECMO in various populations [6]. Coupled with improving ECMO technology, these advances resulted in significant improvements in ECMO utilization, morbidity, and mortality [7].

### 1.2. Modern Applications

With improvements in ECMO technologies, institutional experience, and standardization of use, the applications of ECMO have expanded. These include management of acute severe respiratory failure, as well as use as a bridge to transplantation or support during acute exacerbation of chronic lung diseases such as cystic fibrosis [8,9]. Recently, there has been an evolving role of ECMO support for patients with cardiopulmonary failure due to coronavirus disease 2019 (COVID-19) for both acute respiratory distress syndrome as well as myocarditis and multi-systemic inflammatory syndrome in children (MIS-C). While the data and outcomes for this particular indication are beyond the scope of this review article, conventional selection criteria and management practices should be the standard [10].

Within the pediatric congenital heart disease (CHD) population, ECMO is used to support patients with cardiac or cardiorespiratory failure related to pre-operative hemodynamic instability, post-operative low cardiac output syndrome (LCOS), and inability to wean from CPB [1,11]. In recent years, the rapid implementation of venoarterial (VA) ECMO during cardiopulmonary resuscitation [extracorporeal cardiopulmonary resuscitation (ECPR)] has resulted in improved survival, especially in pediatric patients [6]. This review will focus on the use of ECMO in pediatric and neonatal patients with congenital heart disease.

### 1.3. Trends in Pediatric/Neonatal Cardiac ECMO

Based on the most recent ELSO International Summary, there have been 9620 neonatal and 14,078 pediatric ECMO runs for cardiac failure, comprising 31% of all runs within those age groups [11]. In addition, there have been 7943 (10%) ECPR runs within these age groups. ECPR is the fastest increasing application of ECMO within the neonatal and pediatric age groups. The average survival has remained stable over recent years for patients receiving ECMO for cardiac failure, with 43% of neonatal and 53% of pediatric patients surviving to discharge or transfer. This remains below the average for respiratory failure in both age groups (73% and 60%, respectively) [11]. CHD is the most common indication for cardiac ECMO (81% neonatal, 52% pediatric), with pathologies primarily affecting function (myocarditis, cardiomyopathy) making up most of the remainder [1].

Based on data from the Society of Thoracic Surgeons (STS), MCS is required (ECMO in >95%) in 2.8% of neonatal/pediatric patients in the perioperative setting [12]. Identified risk factors for MCS in this population include young age (average age 13 days) and STS-defined perioperative risk factors (mechanical ventilation, arrhythmia, or shock). Higher complexity operation (STAT category 4 or 5) and prolonged CPB time are also associated with a higher risk of requiring MCS. The highest rates of MCS occur in patients undergoing a Norwood procedure, complex biventricular repairs, and creation of a Damus–Kaye–Stansel (DKS) connection.

## 2. Indications

In the pediatric patient with CHD, ECMO cannulation should be viewed as temporary support or a means to an end, rather than a destination or long-term therapy. There are four general approaches to ECMO cannulation in patients with CHD and cardiac failure [11]. When a patient is expected to have reversible underlying pathology and recovery of cardiac function is anticipated, ECMO can be used as a bridge (temporary support) to recovery. For patients with non-recoverable cardiac disease who are candidates for long-term support, ECMO can be used to bridge to VAD support. ECMO can also be used as a bridge to cardiac transplantation. Finally, in patients with significant comorbidities with variable expectations of recovery, ECMO can be used as a bridge to further decision-making while gathering relevant information or evaluating the progression of comorbidities.

For the patient with CHD, the indication for VA ECMO is almost invariably the development of cardiogenic shock unresponsive to medical therapies. While criteria must be tailored to each patient’s underlying cardiac disease, findings consistent with cardiogenic shock generally include systemic systolic hypotension, oliguria, hyperlactatemia, central venous oxygen desaturation, increased arterial-venous oxygen saturation difference, and altered mental status recalcitrant to maximized medical support [11]. Within the pediatric CHD population, ECMO is most commonly required for pre-operative stabilization, post-operative support, or in the setting of ECPR.

### 2.1. Pre-Operative ECMO Support

Initiation of ECMO is sometimes required in neonates who present with cardiogenic shock related to unpalliated congenital heart disease. While rare, examples of this would include obstructed total anomalous pulmonary venous return, severe Ebstein’s anomaly with a circular shunt, and Tetralogy of Fallot with absent pulmonary valve [13,14,15]. In a study by Mascio and colleagues looking at 96,596 congenital cardiac operations from 80 hospitals, they found 0.5% of patients were supported with mechanical circulatory support pre-operatively (*n* = 463) with another 0.1% (*n* = 151) supported both pre-operatively and post-operatively [12]. In these cases, ECMO is employed to improve end-organ perfusion and provide physiologic stability prior to an operative repair [11].

### 2.2. Post-Operative ECMO Support

The majority of ECMO runs occur in the post-operative setting (2.2% compared to 0.5% pre-operatively and 0.1% both pre- and post-operatively) [12]. The highest rates of MCS have been observed for the Norwood procedure (17.0%), in complex biventricular repairs including the arterial switch operation with a ventricular septal defect and aortic arch repair (14.0%), truncus arteriosus repair (9.4%), and the Ross Konno procedure (9.3%) [12]. Post-operative cardiac dysfunction can manifest in a failure to wean from CPB or development of LCOS in the intensive care unit. Patients requiring ECMO due to failure to wean from CPB are at increased risk for poor outcomes with an in-hospital mortality rate of 55% [16]. Based on a multivariable analysis of this population, the risk of mortality is increased in young patients (<26 days of life) and patients with pre-operative risk factors including prolonged mechanical ventilation, bicarbonate administration, and prior cardiac arrest. Operative factors included complex surgical repairs and long CPB times (>251 min) [16].

LCOS occurs in approximately 10–25% of pediatric patients following cardiac surgery and is characterized by low cardiac output and end-organ hypoperfusion resulting in hyperlactatemia, increased arterial-venous oxygen saturation difference, and oliguria occurring within 24 h after surgery [5,17]. While most patients with LCOS are medically managed, ECMO should be considered for those patients who have evidence of persistent dysfunction and end-organ hypoperfusion despite maximized medical therapies. In these patients, a thorough investigation for residual anatomic defects should be undertaken. In one study, 40% of patients requiring early initiation (within 48 h of surgery) of ECMO after cardiac surgery were found to have a residual defect requiring additional intervention [18]. Indications for cardiac catheterization in this population as well as the timing of ECMO initiation for post-operative cardiac dysfunction are discussed in detail below.

Additional indications for post-operative ECMO support include incessant arrhythmias, such as junctional ectopic tachycardia, resulting in hemodynamic instability [11]. In these patients, an evaluation for residual cardiac lesions or coronary ischemia should be performed. ECMO can provide hemodynamic stability while performing this investigation and gaining control of the patient’s arrhythmia.

### 2.3. ECPR

ECPR is defined as the initiation of ECMO during conventional CPR or within 20 min of return of spontaneous circulation [6]. As with other applications of ECMO, the ultimate goal is to restore effective oxygen delivery and removal of carbon dioxide and to decrease the risk of ischemic-reperfusion injury. Annual cases of ECPR have been increasing within the neonatal and pediatric populations [1]. A large study of pediatric cardiac ECPR cases previously demonstrated a 42% survival rate, slightly lower than the average for all pediatric cardiac ECMO runs [19]. Subsequently, Lasa and colleagues demonstrated improved survival to hospital discharge with favorable neurologic outcomes in patients with ECPR compared with conventional CPR in children after an in-hospital cardiac arrest [20]. When compared to a general population, pediatric patients with CHD have physiologic constraints that limit effective conventional CPR including limited stroke volume, atypical circulation patterns including systemic and pulmonary circulations in series, and limited cerebral perfusion [6,21]. Because of these factors, ECMO team activation should be considered as early as possible during an acute event. ELSO has provided consensus guidelines for the implementation of ECPR in this population [6].

### 2.4. Contraindications

As ECMO technology and experience have increased, the number of absolute contraindications has decreased. ELSO has previously described general contraindications to ECMO support [5]. Specific to neonatal and pediatric cardiac cases, ECMO is generally contraindicated in states of prolonged cardiogenic shock (>6 h), significant prematurity (<32 weeks gestational age), low birth weight (<1.5 kg), severe chromosomal abnormalities, significant brain damage or intracranial hemorrhage (grade III or IV intraventricular hemorrhage), or uncontrollable bleeding [11]. Additionally, a given institution’s experience, outcomes, and availability of staff and resources should be factored into the decision making for ECMO candidacy.

## 3. Timing of ECMO Initiation

There is general agreement that it is best to initiate ECMO prior to the development of severe tissue hypoxia and end-organ dysfunction [10,22]. Numerous studies have demonstrated an association between delayed ECMO initiation and poor outcomes in patients with CHD [22,23,24]. However, the precise optimal timing for initiation of ECMO has not been definitively determined in this population. Studies examining the timing of ECMO initiation in the post-operative period have shown conflicting and inconsistent results [25,26,27,28,29]. Only one study has demonstrated a statistically significant difference in hospital survival depending on whether a patient was cannulated in the operating room or in the intensive care unit [29]. In this case, survival was significantly higher when ECMO was initiated in the operating room (64% vs. 29%, *p* = 0.003). This is consistent with a recent study in adult post-cardiotomy patients with cardiac dysfunction [24]. In this retrospective study, patients empirically initiated on VA ECMO in the operating room were compared to those initiated only after multiple failed CPB weans. The early ECMO group had lower mortality (42% vs. 69%), lower continuous renal replacement therapy rate (67% vs. 89%), improved cardiac index, and more rapid lactate clearance. Ultimately, determining optimal timing to initiate ECMO remains difficult. An approach that takes into account the unique physiology and clinical trajectory of the individual patient is necessary until further studies can supplant this approach.

## 4. Approach to Cannulation

The cannulation site and ECMO strategy for cardiac ECMO are determined by a patient’s underlying cardiac anatomy, prior surgical interventions, and size. VA ECMO can be performed via peripheral cannulation (most commonly drainage via right internal jugular vein, return via right common carotid artery) or central cannulation (most commonly drainage via right atrium, return to ascending aorta). Central cannulation is generally easily accessible and preferable in the early post-cardiotomy period or if peripheral vessel size and cannula selection would be prohibitive to achieving adequate ECMO flow. This is especially important to consider in the setting of shunt-dependent CHD (i.e., hypoplastic left heart syndrome following Norwood procedure with modified Blalock-Taussig shunt [mBTS]). In these patients, the position of the arterial cannula relative to the mBTS and parallel circulations can result in pulmonary overcirculation and inadequate systemic flow. As a result, ECMO flows approaching 200 mL/kg/minute are often required [27,30]. Accordingly, the influence of cannula size needed on cannulation site decision-making cannot be overstated. Further considerations of univentricular physiology are discussed later in this review.

Peripheral cannulation of the neck, while less invasive than central cannulation, has several important limitations. Cannulation utilizing the carotid artery is dependent upon an intact Circle of Willis for adequate perfusion of the ipsilateral cerebral hemisphere. Neurologic complications related to neck cannulation have been reported [31,32]. Groups have demonstrated successful repair and patency of the jugular vein and carotid artery after ECMO decannulation; however, the long-term neurodevelopmental benefit of restoring bilateral carotid flow remains unclear [33,34]. Since the affected jugular vein and carotid artery may not be repaired or deemed unrepairable at the time of decannulation, options for future vascular access and ECMO runs are also limited with this form of peripheral cannulation. In older or larger children, peripheral cannulation using femoral vessels can be considered. In the case of CHD with mixed systemic venous return (i.e., single ventricle anatomy with superior cavopulmonary anastomosis), multiple venous access sites are often required.

Whether due to patient size, anatomy, or inadequate flow, central cannulation is often required for patients with CHD. In the case of inability to wean from CPB, conversion to central ECMO is simple [16]. Central cannulation generally allows the use of larger cannulas than peripheral cannulation with associated higher maximum flow rates. Central cannulation is a more invasive procedure and requires a specialized team with expertise in this cannulation approach. In the post-operative setting, one of the most common complications is significant bleeding [35]. Other limitations include the risk of dislodgement of cannulas, the need for repeat mediastinal exploration, and large vessel injury [30].

In the ECPR setting, cannulation can be either central or peripheral and should be tailored to the patient’s anatomy, size, clinical factors, including previous sternotomies, and recent cardiac operations. One study demonstrated a higher survival rate with peripheral cannulation in pediatric ECPR cases, though this may be confounded by a higher prevalence of non-operative patients in the peripheral cannulation group [19]. At present, there is no clear consensus on which approach is superior for patients with CHD [6]. In small children without recent sternotomy, the right-sided carotid artery and jugular vein may be faster to cannulate than performing sternotomy and cannulating the central vessels. The presence of a recent sternotomy, complex cardiac anatomy, peripheral vessel stenosis or occlusion, or the need for larger cannulas makes central cannulation preferable for some patients. ECPR with central cannulation requires periods of internal cardiac massage which may complicate cannulation, preclude consistent compressions, and requires a different skill set than standard chest compressions. More data to determine optimal cannulation technique in ECPR is warranted.

## 5. Cardiac Catheterization

As previously mentioned, it is critical to evaluate for residual lesions in those who require ECMO for post-cardiotomy support. It has been shown that cardiac catheterization can be safely performed in children on ECMO [36,37]. Booth and colleagues described their experience at a single, large tertiary children’s hospital. At the time of the study, 28% of cardiac patients on ECMO underwent cardiac catheterizations, with 83% of those patients undergoing an intervention either during the catheterization or subsequently in the operating room [36]. More recently, another single, large tertiary children’s hospital published their data on a total of 55 catheterizations performed on 51 patients during 53 unique cardiac ECMO courses. Again, 83% had either surgical or catheter intervention at the time of or following the procedure. High severity complications occurred in three (5.6%) of the patients, including one death due to hemothorax after pulmonary artery stent placement [37]. This is comparable to the overall high severity adverse events in congenital cardiac catheterization, which has been reported to occur in 9% of interventional cases and 5% of diagnostic cases [38].

The importance of addressing residual lesions has been described, as well as the safety of catheter-based evaluations on ECMO, but the question remains of when to proceed. Non-invasive evaluations such as echocardiography may aid in identifying a need for catheter or surgical-based interventions; however, they should not delay invasive evaluation. Earlier performance of catheterization on ECMO in this cohort of patients is associated with better survival [39,40]. Thus, despite the known risk of procedural complications in these critically ill patients, early identification and intervention of residual lesions are paramount.

## 6. Unloading the Ventricle and Left Atrial Decompression

Left atrial decompression is a protective factor for mortality, and decompression of the left heart is essential for myocardial recovery [16,41]. ECMO does not decompress the left ventricle, and there is always blood returning to the left atrium from the thebesian and bronchial veins despite optimal ECMO flow. In the setting of cardiac dysfunction, draining the left atrium can reduce left ventricular distension allowing for myocardial rest and recovery. In addition, left atrial decompression protects the lungs from injury due to resultant pulmonary edema from severe left atrial hypertension [16,42]. The optimal method of left atrial decompression has not been determined, and various methods exist both surgically and via the percutaneous method. These include placement of a left atrial vent, balloon atrial septostomy, and atrial septal stent placement [42]. Determination of which method to use should depend on patient-specific factors and surgical/catheterization resources at individual institutions until further studies are performed.

Earlier time to decompression following ECMO cannulation is associated with a shorter duration of ECMO and ICU length of stay [43,44]. It is important to note that an increased odds of survival has not been shown in children with congenital heart disease; however, survival benefit has been shown in those with myocarditis or dilated cardiomyopathy [41,43,45]. This is likely multifactorial in nature and requires additional study in patients specific to congenital heart disease. Nevertheless, given the known morbidities associated with prolonged ECMO use, early left atrial decompression is justified. Specifically, Zampi and colleagues found late left atrial decompression, defined as ≥18 h, was associated with a longer duration of ECMO support and mechanical ventilation (although no survival benefit was demonstrated), and so efforts to intervene in that time period are appropriate [43].

ECMO does decompress the right side of the heart. Thus, patients on ECMO with primarily right ventricular dysfunction, such as those on ECMO for pulmonary hypertension, with adequate left ventricular dysfunction, may not require left atrial decompression.

## 7. Circuit Considerations

Given the majority of patients with congenital heart disease on ECMO are neonates, there are some neonatal circuit considerations to discuss. First, the consideration of roller versus centrifugal pumps. While there was a recent propensity-matched study by O’Halloran and colleagues on the comparison of the two pump types in neonates, with better outcomes in the roller pump group, there are several limitations to extrapolating this data to our cohort [46]. Mainly, the time period in which the study was performed. In the earlier era of the study, the pump type was predominately roller and predominately centrifugal in the latter part of the study. This is relevant to the known learning curve with new technology regarding mechanical circulatory support within institutions. In addition, the poorer outcomes were attributable to hemolysis, which was not able to be well-defined in the study given registry limitations. Further analysis on pump type specific to neonates with cardiac disease might better inform us.

There are additional challenges of cannula size and position in neonates. In the setting of smaller vessels and cannulation options, issues of venous drainage (if the venous cannula is too small) can result in the inability to achieve the desired venous return. Furthermore, higher arterial pressure (if the arterial cannula is too small or malpositioned), leading to higher post-membrane pressure and thus hemolysis, may occur [47]. Lastly, with regards to priming volume, this often serves as an entire exchange transfusion in a neonate. Important considerations of the priming volume for the circuit in relation to the total blood volume in a neonate are the risks of high potassium, low calcium, and fluid shifts [47]. Recognition of the impact of high priming volume on neonates during cannulation by the provider is imperative.

## 8. Anticoagulation

Anticoagulation remains a challenge given the need to prevent circuit thrombosis while minimizing the risk of bleeding to the patient. This is particularly challenging in neonates given their immature coagulation system, lower circuit flow, and higher risk of cerebrovascular accidents [48]. Traditionally, heparin has been the mainstay of anticoagulation. Heparin has advantages of familiarity among providers, reversibility, and low cost. However, the pharmacokinetic effect is variable and relies on antithrombin III.

Recent studies have shown promising results with direct thrombin inhibitors such as bivalirudin, but experience in the pediatric population on ECMO is limited. Hamzah and colleagues reported an analysis of pediatric patients on ECMO comparing 16 anticoagulated with bivalirudin and 16 with heparin and found less bleeding in the bivalirudin group [49]. Other outcomes including thrombotic risk were not significantly different between the two groups. Thus, the authors concluded that bivalirudin was feasible, safe, reliable, and cost-effective in comparison to heparin. Many institutions have transitioned their primary anticoagulant of pediatric patients on ECMO to bivalirudin, and so further collaboration and data sharing are possible. Much of the knowledge we have on the use of bivalirudin for ECMO in pediatrics is based on the experience from our ventricular assist device (VAD) colleagues for the berlin EXCOR [50]. Harmonization documents developed by the Advanced Cardiac Therapies Improving Outcomes Network (ACTION), a network for children with heart failure and those being supported with a VAD, have taught us a great deal in management, but experience specific to ECMO in children with congenital heart disease is imperative [51].

Monitoring of anticoagulation in children is fraught with its own challenges. Specific to children with congenital heart disease on ECMO, they are often neonates with immature coagulation systems and may have disease processes that lend to protein and factor losses further complicating the clinical picture. Heparin is monitored by activated clotting time (ACT), activated partial thromboplastin time (aPTT), or anti-factor Xa (anto-FXa) levels, though ACT is poorly correlated with anti-FXa and antithrombin III modifies the relationship between ACT and heparin dose. There are also age-related differences in the reliability of those tests [52]. Bivalirudin anticoagulation effect is measured by aPTT and dilute thrombin time; however, the dose of bivalirudin needs to be increased over time in order to maintain aPTT within the therapeutic range. It is important to note that the international normalized ratio (INR) will be prolonged by bivalirudin as well. Rotational thromboelastometry (ROTEM) has been shown to be a useful tool in evaluating the anticoagulant effect of bivalirudin in children on ECMO, but more experience and clinical data are needed [53]. Regardless, the limitation in anticoagulation management of these patients is not necessarily the medications, but rather a need for better monitoring methods and continued collaboration.

## 9. Considerations in the Univentricular Heart

Historically, functionally univentricular circulations were considered a contraindication to ECMO, as early reports highlighted a significantly increased risk of mortality for patients with single ventricle (SV) physiology [19,54]. However, the practice has grown more prevalent over the last several decades with evidence of the potential benefits of ECMO as a therapeutic rescue. In the examination of ECMO utilization following the first stage of reconstruction for hypoplastic left heart syndrome and its variants, Ravishankar et al. demonstrated almost 40% of their cohort survived to hospital discharge with all patients who experienced shunt thrombosis surviving to discharge [55]. Furthermore, a review of infants with shunted SV physiology supported with VA-ECMO at Children’s Hospital Boston between 1996 and 2005 found overall survival to discharge in this group of neonates (48%) equivalent to all neonatal and pediatric cardiac VA-ECMO in the Extracorporeal Life Support Organization registry (41%) [56]. Accordingly, ECMO support has come to be used in single-ventricle patients for cardiopulmonary failure in the setting of failure to separate from cardiopulmonary bypass, refractory low cardiac output syndrome, acute shunt thrombosis, hemodynamically unstable arrhythmias, and as a part of ECPR. In fact, ECMO usage and outcomes among congenital heart surgical patients as documented in the Society of Thoracic Surgeons Database demonstrated patients after the Norwood operation had the highest rate of post-operative mechanical support (17%) [12].

A caveat to this increased utilization is that while the use of ECMO in the SV patient population has steadily increased, mortality has remained stagnant between 50–60% for the last several decades [57]. The variability in survival for infants with shunted functionally univentricular circulations is multifactorial, but data from several studies suggest patients cannulated for shunt thrombus and hypoxia have better outcomes in terms of survival compared with those supported for ventricular dysfunction or after cardiac arrest [56].

### 9.1. Systemic to Pulmonary Shunt

Another relevant consideration for outcomes of this population is the type of systemic to pulmonary shunt in place. With patients palliated with the modified Blalock-Taussig shunt, hesitancy to utilize ECMO was based on the potential limitation to adequate support of systemic perfusion created by the shunt runoff. Attempts to provide a balanced circulation effectively on mechanical support have focused on limiting the pulmonary blood flow via shunt clipping increased ECMO output with a patent shunt at the risk of pulmonary overcirculation. Early attempts to occlude the shunt surgically at the time of ECMO in 4 patients but had no survivors in this group. In contrast, in the 5 patients in whom the shunt was left patent and the ECMO flow increased to compensate for shunt runoff, mortality was considerably lower, at 20% [27]. A later study evaluated 44 patients who underwent ECMO following single-ventricle palliation with an mBT shunt where those patients demonstrating compromised systemic perfusion with signs of poor end-organ perfusion had surgical clips placed to restrict the shunt diameter, reduce runoff and lead to improvement in survival to hospital discharge [56]. An alternative strategy that can be employed is to augment overall output with significantly higher ECMO flows without surgical reduction of shunt diameter. In one series, survival to discharge in all infants and neonates requiring ECMO was comparable to that reported in previous studies and was not affected by the presence of a mBTS [58]. This same study also highlighted that SV patients with an RV-PA conduit as their systemic to pulmonary shunt were less prone to runoff because of ventricular decompression on VA ECMO, thus less frequently requiring increased ECMO flows or clipping of the shunt. Additionally, those with Sano shunts may benefit from some ejection of the ventricle through the Sano to avoid stasis and subsequent occlusion of the shunt itself.

### 9.2. Superior Cavopulmonary Anastomosis

MCS following subsequent stages of single ventricle palliation also presents unique challenges that directly impact the technical approach and overall outcomes. The Glenn procedure (superior cavo-pulmonary connection) results in the separation of the superior and inferior caval circulations, and thus careful consideration must be given to cannulation strategy. Knowledge of the patient’s specific venous anatomy including right, left, or bilateral SVCs as well as vessel patency is imperative. Furthermore, successful mechanical support relies on adequate system venous drainage. As the SVC is now anastomosed to the pulmonary artery, decompression of the systemic pumping chamber via IVC cannula may not guarantee adequate venous drainage from that circulation. To achieve adequate decompression in the setting of elevated Glenn pressure, whether secondary anatomic obstruction, elevated pulmonary vascular resistance, pulmonary vein stenosis, restriction at the atrial septum, elevated ventricular end-diastolic pressure, or depressed ventricular function, a second venous cannula in the SVC is often required [59]. Outcomes for patients with bidirectional Glenn physiology have been historically very poor, particularly in the setting of eCPR. This population is exceedingly susceptible to neurologic injury, including seizures, hemorrhage, and embolic stroke [31]. High cavopulmonary pressures and thus elevated cerebral venous pressures in combination with low systemic blood pressure results in decreased cerebral perfusion pressure and likely predisposes these patients to neurologic injury, especially in the setting of cardiopulmonary resuscitation. Some centers have advocated for cannulation of the SVC/Glenn circulation prior to IVC cannulation, but little data exists to support this as preventive to neurologic injury [60].

## 10. Weaning ECMO Support

ECMO can be an effective rescue therapy for cardiopulmonary failure yet is not a long-term strategy of mechanical support in part due to the potential complications associated with its use. While indications for ECMO utilization are abundant in the literature, little consensus exists regarding how to wean ECMO support. To further complicate matters, as the reasons for initiating veno-venous (VV) vs veno-arterial (VA) ECMO initiation are very different, so are the parameters and approach to weaning support. For either type of support, there are common themes. Given the underlying goal of supporting end-organ function and allowing time for recovery, diligence is necessary to prevent secondary injury and for organ function to return. Neurologic injury is a common complication among patients supported with ECMO and is associated with significant mortality and long-term morbidity [61]. Frequent neurologic assessment, screening head imaging, close anticoagulation monitoring, and avoidance of neuromuscular blockade are essential to prevent, or at least guarantee early recognition, of neurologic injury. Acute kidney injury (AKI) and fluid overload occur frequently on ECMO and have been identified in numerous reports to be independently associated with increased mortality [62,63].

In VV ECMO, as its purpose is to support gas exchange in the setting of respiratory failure, the injured lung must be allowed to recover by minimizing further trauma from mechanical ventilation. To address this issue, early extubation has been increasingly suggested to avoid the risk of ventilation-induced lung injury, ventilator-associated pneumonia, and exacerbation of the initial lung injury. However, early reports suggest that this strategy is still limited to a small subset of patients [64]. The process of weaning can only commence once the lungs recuperate, marked by improving compliance and better gas exchange. Accordingly, the duration of VV ECMO therapy can be numbered in days to months depending on the etiology. Once clinical improvement on chest radiograph and adequate tidal volumes are achieved on reasonable ventilator settings (PIP < 27 cmH2O, PEEP < 12 cmH2O), ECMO flow is progressively reduced, followed by tapering the sweep gas and FiO2. Some centers have advocated for maintaining stable blood flows to prevent any additional clotting, others reduce the blood flow to a minimum to reduce shear stress for the circulating blood [65].

In contrast, weaning from VA ECMO support for cardiopulmonary failure is primarily dependent on cardiac contractile recovery. Attempts to wean rarely occur in the first 48–72 h, to allow ample time for myocardial recovery and end-organ function. Compared to VV ECMO, the duration of VA ECMO is shorter with an average duration of support of 3–8 days across various studies and is rare to last more than 1–2 weeks [66]. Whatever the inciting cause is (post-operative dysfunction, myocarditis, or cardiomyopathy, arrhythmia, drug intoxication, or sepsis-induced dysfunction); VA ECMO support should remain in place until the patient has recovered from that insult. In addition, patients should demonstrate adequate pulmonary compliance and be able to maintain an appropriate oxygen saturation with an FiO2 on the ventilator of less than 60% and maintain hemodynamically acceptable blood pressures with a measurable pulse pressure on nontoxic vasoactive support. Weaning trials can be informative in assessing how decreased ECMO flows, and the associated increase in preload, affect not only left and right ventricular function, but also the ventricular-ventricular interaction [67,68]. Numerous quantitative and qualitative parameters have been described in the literature as being strong predictors of successful weaning. Most weaning protocols utilize clinical hemodynamic assessments with mean blood pressure and pulse pressure, as well physiologic biomarkers such as lactate or B-type natriuretic peptide [68,69,70]. Current data suggest, however, echocardiography represents the best tool to determine ventricular recovery. Interestingly, assessments of systolic LV function (LV ejection fraction, aortic velocity-time integral, and lateral mitral annulus peak systolic velocity) have been associated with successful weaning, while measurements of diastolic (mitral inflow velocities and ratios) do not [71]. Unfortunately, despite progress in elucidating positive predictive parameters, weaning and separating from mechanical support do not always equate to survival, as a significant percentage of patients successfully separated from ECMO do not survive to hospital discharge.

## 11. Short-Term Outcomes

Overall survival throughout the past three decades has been 59–61% in neonatal and pediatric ECMO [1]. Mortality associated with post-operative MCS in children with congenital heart disease is 53%, compared to those who do not undergo post-operative MCS, who have an overall mortality rate of 2.9% [12]. However, it is important to recognize that likely few, if any, of those patients would have been discharged alive without MCS support. Additionally, though the survival rates have remained relatively static, the acuity and underlying etiology of the need for support of patients being cannulated to ECMO have only expanded.

Neurologic complications are common in children on ECMO support, particularly in those who have undergone ECPR [1]. Of complications reported to ELSO from 2015–2020 in pediatric patients who underwent ECPR, 7.4% met brain death criteria. Pediatric patients who did not undergo ECPR and were on ECMO for either respiratory or cardiac indications had a lower rate of brain death of 1.8–2%. Brain death is less commonly reported in the neonatal population; 0.2–0.7% for all ECMO indications [5]. Renal dysfunction is another common complication of children and neonates on ECMO. At least 40% of pediatric patients have renal dysfunction, defined by the ELSO registry as creatinine > 1.5 or the requirement of renal replacement therapy. Specifically, 29.7–32.2% of pediatric patients undergo renal replacement therapy while on ECMO, which is similar to the neonatal cohort of 27–36% [5].

## 12. Long-Term Outcomes of ECMO Survivors

As the utilization of ECMO to support children with cardiopulmonary failure has become more frequent, there has been increasing awareness that survival to discharge is a poor metric for success. In the past decade, there has been more investigation into long-term outcomes following ECMO support, particularly regarding chronic care requirements, neurodevelopmental milestones, and quality of life indices. The risk of chronic respiratory failure in patients who survive ECMO is not negligible. In one series, pediatric patients supported with VV ECMO had a 14% rate for tracheostomy placement and a 9% rate of mechanical ventilation at discharge [72]. Neurologic injury consisting of seizures, intracranial hemorrhage, and stroke are an unfortunate and frequent complication seen in ECMO survivors, up to 20%, and associated with significant long-term functional impairment [73]. Cardiac ECMO patients in one study demonstrated neurodevelopmental scores at least one standard deviation below the normative mean in the gross motor, language, and cognitive domains as compared to matched non-ECMO controls [74]. Other investigations have interestingly drawn distinctions between VA ECMO and other methods of mechanical circulatory support and found that while 60% of children supported with ECMO had moderate to severe neurologic impairment, only 20% of those surviving after ventricular assist device support demonstrated the same degree of neurologic impairment [75]. Whether due to decreased requirements for anticoagulation with ventricular assist devices or the tendency of that subset of patients to be older and at less risk, the cause for this discrepancy remains unclear.

There has also been increasing emphasis on categorizing long-term psychosocial ramifications of surviving critical illness requiring ECMO support. One assessment of the quality of life demonstrated that pediatric cardiac ECMO survivors were reported by parental response to have lower quality of life scores compared to both age-matched controls and other cardiac populations [76]. These lower scores were primarily driven by the physical component of health-related quality of life score, while the reported psychosocial quality of life was similar to that of the general population and of other pediatric cardiac populations. Another single-center study demonstrated that the majority of pediatric cardiac ECMO survivors reported positive outcomes with respect to health and physical limitations, but children reported lower quality of life compared with healthy individual normative data [77]. Taken together, it is abundantly clear that more structured and long-term support must be assembled to support the developmental function and quality of life in this population.

## 13. Conclusions

Mechanical circulatory support has grown over the last several decades to become a vital therapy in children refractory to maximal medical management. Extracorporeal membrane oxygenation can now be quickly and efficiently deployed to support critically ill patients with cardiopulmonary failure of diverse etiologies. The field has grown in part due to advances in technology and techniques, but also from continued growth in the expertise to successfully manage these patients. It is sobering to recognize that mortality for pediatric patients placed on ECMO support remains high, and of those who are able to successfully separate, some will not survive until discharge, while others will continue to be burdened by complications accrued while critically ill on mechanical support. Collaboration and research specific to ECMO in CHD patients are imperative to improving outcomes.

## Data Availability

Not acpplicable.

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
