# Peer review of "Extracorporeal Membrane Oxygenation in Congenital Heart Disease"

_children, 2022, doi:10.3390/children9030380_

Round 1

Reviewer 1 Report

In the present work (Extracorporeal Membrane Oxygenation in Congenital Heart Disease), authors expose an interesting review about the usefulness of Extra-corporeal Membrane Oxygenation (ECMO) in the clinical setting of Congenital Heart Disease. It is really well written, it is concise and it provides some up-to-date information in this setting.

Although authors have written a really interesting review, there are still some issues that can be improved.

#1. In the review is mentioned that about 2.8% of pediatric patients require ECMO in the perioperative period. In this sense, authors should include some references to the surgeries most related with ECMO requirements.

#2. When talking about indications, authors should support the review with some data about the relative number of patients in each group (mainly comparing between preoperative and postoperative indications). Moreover, when talking about the different scenarios in which postoperative ECMO is implantated, authors should include the relative number of each scenario.

#3. In the introduction section authors expose some modern applications of ECMO. They describe superficially its usefulness in several respiratory failure. In this sense, authors should include some references to the current indication of ECMO in respiratory failure due to Covid infection.

Author Response

Reviewer 1:

Thank you so much to the reviewers for your thoughtful comments and questions. We do feel the revisions based on your recommendations make our manuscript stronger, and we appreciate the time you gave to improving our work. Our responses are written below.

In the present work (Extracorporeal Membrane Oxygenation in Congenital Heart Disease), authors expose an interesting review about the usefulness of Extra-corporeal Membrane Oxygenation (ECMO) in the clinical setting of Congenital Heart Disease. It is really well written, it is concise and it provides some up-to-date information in this setting.

Although authors have written a really interesting review, there are still some issues that can be improved.

#1. In the review is mentioned that about 2.8% of pediatric patients require ECMO in the perioperative period. In this sense, authors should include some references to the surgeries most related with ECMO requirements.

Response: Thank you for this observation. We agree the most common surgeries requiring ECMO is an important point, and this we do include in the text,Identified risk factors for MCS in this population include young age (average age 13 days) and STS-defined perioperative risk factors (mechanical ventilation, arrhythmia, or shock). Higher complexity operation (STAT category 4 or 5) and prolonged CPB time are also associated with higher risk of requiring MCS. The highest rates of MCS occur in patients undergoing a Norwood procedure, complex biventricular repairs, and creation of a Damus-Kaye-Stansel (DKS) connection.” We additionally elaborate further, which is described in response to your second comment.

#2. When talking about indications, authors should support the review with some data about the relative number of patients in each group (mainly comparing between preoperative and postoperative indications). Moreover, when talking about the different scenarios in which postoperative ECMO is implantated, authors should include the relative number of each scenario.

Response: We agree this is an important addition to this manuscript, and have added the following, “a study by Mascio and colleagues looking at 96,596 congenital cardiac operations from 80 hospitals, they found 0.5% of patients were supported with mechanical circulatory support pre-operatively (n=463) with another 0.1% (n=151) supported both pre-operatively and post-operatively [12].

The majority of ECMO runs occur in the post-operative setting (2.2% compared to 0.5% pre-operatively and 0.1% both pre and post-operatively) [12]. The highest rates of MCS have been observed for the Norwood procedure (17.0%), in complex biventricular repairs including the arterial switch operation with a ventricular septal defect and aortic arch repair (14.0%), truncus arteriosus repair (9.4%), and the Ross Konno procedure (9.3%) [12].”

#3. In the introduction section authors expose some modern applications of ECMO. They describe superficially its usefulness in several respiratory failure. In this sense, authors should include some references to the current indication of ECMO in respiratory failure due to Covid infection.

Response: Thank you, this is an excellent point and important given the timing of the paper being written. We have added to out modern applications section, “Recently, there has been an evolving role of ECMO support for patients with cardiopulmonary failure due to coronavirus disease 2019 (COVID-19) for both acute respiratory distress syndrome as well as myocarditis and multi-systemic inflammatory syndrome in children (MIS-C). While the data and outcomes for this particular indication is beyond the scope of this review article, conventional selection criteria and management practices should be the standard [10].”

Reviewer 2 Report

The authors of this manuscript aim to undertake a comprehensive review of extracorporeal membrane oxygenation support use in children with congenital heart disease. It is important to distinguish pediatric patients with congenital heart disease from other pediatric patients requiring ECMO support as often the indications, cannulation techniques, management strategies, and outcomes can be different. To that end, this review provides a valid contribution to the literature. The review is comprehensive, and previously published data and studies are well cited. The writing is clear, and the section headings are appropriate.

I would suggest including a section on short-term outcomes for the patients cannulated to ECMO in addition to the long-term outcomes section, particularly with regard to renal dysfunction and need for CRRT.

While I do think it is important to note in your conclusion that mortality is still high, as the authors note earlier in the manuscript, the acuity and underlying etiology of need for support of patients being cannulated to ECMO has only broadened which likely plays a role in survival appearing stagnant, and this should be noted.

Author Response

Reviewer 2

Thank you so much to the reviewers for your thoughtful comments and questions. We do feel the revisions based on your recommendations make our manuscript stronger, and we appreciate the time you gave to improving our work. Our responses are written below.

The authors of this manuscript aim to undertake a comprehensive review of extracorporeal membrane oxygenation support use in children with congenital heart disease. It is important to distinguish pediatric patients with congenital heart disease from other pediatric patients requiring ECMO support as often the indications, cannulation techniques, management strategies, and outcomes can be different. To that end, this review provides a valid contribution to the literature. The review is comprehensive, and previously published data and studies are well cited. The writing is clear, and the section headings are appropriate.

I would suggest including a section on short-term outcomes for the patients cannulated to ECMO in addition to the long-term outcomes section, particularly with regard to renal dysfunction and need for CRRT.

Response: Thank you for this observation. We have added short-term outcomes section to the paper, and described some of the important complications that you mentioned, which now includes, “Neurologic complications are common in children on ECMO support, particularly in those who have undergone ECPR [1]. Of complications reported to ELSO from 2015-2020 in pediatric patients who underwent ECPR, 7.4% met brain death criteria. Pediatric patients who did not undergo ECPR and were on ECMO for either respiratory or cardiac indications had a lower rate of brain death of 1.8-2%. Brain death is less commonly reported in the neonatal population; 0.2-0.7% for all ECMO indications [5]. Renal dysfunction is another common complication of children and neonates on ECMO. At least 40% of pediatric patients have renal dysfunction, defined by the ELSO registry as creatinine >1.5 or the requirement of renal replacement therapy. Specifically, 29.7-32.2% of pediatric patients undergo renal replacement therapy while on ECMO, which is similar to the neonatal cohort of 27-36% [5].”

While I do think it is important to note in your conclusion that mortality is still high, as the authors note earlier in the manuscript, the acuity and underlying etiology of need for support of patients being cannulated to ECMO has only broadened which likely plays a role in survival appearing stagnant, and this should be noted.

Response: Thank you, this is an excellent point, and one that we have now included in the latter portion of this paragraph in our short-term outcomes section. “Overall survival throughout the past three decades has been 59-61% in neonatal and pediatric ECMO [1]. Mortality associated with post-operative MCS in children with congenital heart disease is 53%, compared to those who do not undergo post-operative MCS, who have an overall mortality rate of 2.9% [11]. However, it is important to recognize that likely few, if any, of those patients would have been discharged alive without MCS support. Additionally, though the survival rates have remained relatively static, the acuity and underlying etiology of need for support of patients being cannulated to ECMO has only expanded.”